# Scaling carbon fluxes from eddy covariance sites to globe: Synthesis and evaluation of the FLUXCOM approach

Martin Jung[1], Christopher Schwalm[2], Mirco Migliavacca[1], Sophia Walther[1], Gustau Camps-Valls[3], Sujan Koirala[1], Peter Anthoni[4], Simon Besnard[1,5], Paul Bodesheim[1,6], Nuno Carvalhais[1,7], Frédéric Chevallier[8], Fabian Gans[1], Daniel S. Goll[9], Vanessa Haverd[10], Philipp Koehler[11], Kazuhito Ichii[12,13], Atul K. Jain[14], Junzhi Liu[1,15], Danica Lombardozzi[16], Julia E.M.S. Nabel[17], Jacob A. Nelson[1], Michael O'Sullivan[18], Martijn Pallandt[19], Dario Papale[20,21], Wouter Peters[22], Julia Pongratz[23,17], Christian Rödenbeck[19], Stephen Sitch[18], Gianluca Tramontana[20,3], Anthony Walker[24], Ulrich Weber[1], Markus Reichstein[1]

[1]Department of Biogeochmical Integration, Max Planck Institute for Biogeochemistry, Jena, 07745, Germany
[2]Woods Hole Research Center, Falmouth, MA, 02540-1644, USA
[3]Image Processing Laboratory (IPL), Universitat de València, Paterna, 46980, Spain
[4]Institute of Meteorology and Climate Research – Atmospheric Environmental Research (IMK-IFU), Karlsruhe Institute of Technology, Garmisch-Partenkirchen, 82467, Germany
[5]Laboratory of Geo-Information Science and Remote Sensing, Wageningen University and Research, Wageningen, 6708 PB, Netherlands
[6]Department of Mathematics and Computer Science, Friedrich-Schiller Universität Jena, Jena, 07743, Germany
[7]Departamento de Ciências e Engenharia do Ambiente (DCEA), Faculdade de Ciências e Tecnologia, FCT, Universidade Nova de Lisboa, Caparica, 2829-516, Portugal
[8]Laboratoire des Sciences du Climat et de l'Environnement (LSCE/IPSL), Université Paris-Saclay, Gif-sur-Yvette, F-91198, France
[9]Department of Geography, University of Augsburg, Augsburg, 86159, Germany
[10]Department Continental Biogeochemical Cycles, CSIRO Oceans and Atmosphere, Canberra, 2601, Australia
[11]Division of Geological and Planetary Sciences, California Institute of Technology, Pasadena, USA

[12]Center for Environmental Remote Sensing (CEReS), Chiba University, Chiba, 263-8522, Japan
[13]Center for Global Environmental Research, National Institute for Environmental Studies, Tsukuba, 305-8506, Japan
[14]Department of Atmospheric Science, University of Illinois, Urbana, IL 61801, USA
[15]School of Geography, Nanjing Normal University, Nanjing, 210023, China
[16]Climate and Global Dynamics Laboratory, National Center for Atmospheric Research, Boulder, CO 80307, USA
[17]Department Land in the Earth System (LES), Max Planck Institute for Meteorology, Hamburg, 20146, Germany
[18]College of Life and Environmental Sciences, University of Exeter, Exeter, EX4 4QE, UK
[19]Department of Biogeochmical Systems, Max Planck Institute for Biogeochemistry, Jena, 07745, Germany
[20]Department of Innovation in Biology, Agri-food and Forest systems (DIBAF), University of Tuscia, Viterbo, 01100, Italy
[21]Impacts on Agriculture, Forests and Ecosystem Services (IAFES), EuroMediterranean Center on Climate Change (CMCC), Lecce, 01100, Italy
[22]Department of Meteorology and Air Quality, Wageningen University and Research, Wageningen, 6700 AA, Netherlands
[23]Department of Geography, Ludwig-Maximilans-Universität München, München, 80333, Germany
[24]Climate Change Science Institute, Oak Ridge National Laboratory, Oak Ridge, USA

*Correspondence to*: Martin Jung (mjung@bgc-jena.mpg.de)

**Abstract.** FLUXNET comprises globally-distributed eddy covariance-based estimates of carbon fluxes between the biosphere and the atmosphere. Since eddy covariance flux towers have a relatively small footprint and are distributed unevenly across the world, upscaling the observations is necessary to obtain global-scale estimates of biosphere-atmosphere exchange. Based on cross-consistency checks with atmospheric inversions, sun-induced fluorescence (SIF) and dynamic global vegetation models (DGVM), we provide here a systematic assessment of the latest upscaling efforts for gross primary production (GPP) and net ecosystem exchange (NEE) of the FLUXCOM initiative, where different machine learning methods, forcing datasets, and sets of predictor variables were employed.

Spatial patterns of mean GPP are consistent across FLUXCOM and DGVM ensembles ($R^2$>0.94 at 1° spatial resolution) while the majority of DGVMs show, for 70% of the land surface, values outside the FLUXCOM range. Global mean GPP magnitudes for 2008-2010 from FLUXCOM members vary within 106 and 130 PgC yr[-1] with the largest uncertainty in the tropics. Seasonal variations of independent SIF estimates agree better with FLUXCOM GPP (mean global pixel-wise $R^2$ ~ 0.75) than with GPP from DGVMs (mean global pixel-wise $R^2$ ~ 0.6). Seasonal variations of FLUXCOM NEE show good consistency with atmospheric inversion-based net land carbon fluxes, particularly for temperate and boreal regions ($R^2$>0.92). Interannual variability of global NEE in FLUXCOM is underestimated compared to inversions and DGVMs. The FLUXCOM version which uses also meteorological inputs shows a strong co-variation of interannual patterns with inversions ($R^2$=0.87 for 2001-2010). Mean regional NEE from FLUXCOM shows larger uptake than inversion and DGVM-based estimates, particularly in the tropics with discrepancies of up to several hundred gC m[2] yr[-1]. These discrepancies can only partly be reconciled by carbon loss pathways that are implicit in inversions but not captured by the flux tower measurements such as carbon emissions from fires and water bodies. We hypothesize that a combination of systematic biases in the underlying eddy covariance data, in particular in tall tropical forests, and a lack of site-history effects on NEE in FLUXCOM are likely responsible for the too strong tropical carbon sink estimated by FLUXCOM. Furthermore, as FLUXCOM does not account for $CO_2$ fertilization effects carbon flux trends are not realistic. Overall, current FLUXCOM estimates of mean annual and seasonal cycles of GPP as well as seasonal NEE variations provide useful constraints of global carbon cycling, while interannual variability patterns from FLUXCOM are valuable but require cautious interpretation. Exploring the diversity of Earth Observation data and of machine learning concepts along with improved quality and quantity of flux tower measurements will facilitate further improvements of the FLUXCOM approach overall.

## 1 Introduction

Upscaling local eddy covariance (EC) measurements (Baldocchi et al., 2001) from tower footprint to global wall-to-wall maps uses globally-available predictor variables such as satellite remote sensing and meteorological data (Jung et al., 2011). This forcing data is first used to establish empirical models for fluxes of interest at site level, and then to estimate gridded fluxes by applying these models across all vegetated grid cells. Previous FLUXNET upscaling efforts using machine learning techniques (Beer et al., 2010; Jung et al., 2009; Jung et al., 2011) yielded global products that present a data-driven 'bottom-up' perspective on carbon fluxes between the biosphere and the atmosphere. These 'bottom-up' products are complementary to process-based model simulations and 'top-down' atmospheric inversions. However, estimates of carbon fluxes are subject to uncertainty from choice of machine learning algorithm and predictor variables, forcing data, FLUXNET

measurements and incomplete representation of the different ecosystems therein. The FLUXCOM initiative (www.fluxcom.org) aims to improve our understanding of the multiple sources and facets of uncertainties in empirical upscaling and, ultimately, to provide an ensemble of machine learning-based global flux products to the scientific community. Within FLUXCOM an intercomparison was conducted for two complementary experimental setups of input drivers and resulting global gridded products. These setups systematically vary machine learning and flux partitioning methods as well as forcing datasets to separate measured net ecosystem exchange (NEE) into gross primary productivity (GPP) and Terrestrial Ecosystem Respiration (TER) (Jung et al., 2019; Tramontana et al., 2016).

Evaluating the strengths and weaknesses of the FLUXCOM products and the approaches used therein is crucial to inform potential scientific uses, and to guide future methodological developments. An evaluation based on site-level cross-validation analysis (Tramontana et al., 2016) showed a general high consistency among machine learning algorithms, experimental setups and flux partitioning methods applied in FLUXCOM. However, the conclusions from site-level cross-validation may be limited by potential systematic measurement errors that are inherent in the underlying EC measurements (e.g. Aubinet et al., 2012), or the spatially biased distribution of FLUXNET sites (Papale et al., 2015). Therefore, cross-consistency checks of the FLUXCOM products with independent estimates are important to consider. But such checks are complex due to limitations of the independent approaches or the lack of comparability of similar but not identical variables. In this study, we contextualize FLUXCOM products in relation to independent state-of-the-art estimates of carbon cycling. The comparison strategy prioritises robust features of the independent datasets, and discusses residual uncertainties.

The objectives of this paper are (1) to present a synthesis and evaluation of FLUXCOM ensembles for GPP and NEE against patterns of remotely sensed sun induced fluorescence (SIF) and atmospheric inversion results respectively, (2) to discuss limitations of FLUXCOM and synthesize lessons learned, and (3) to outline potential future paths of FLUXCOM development. Due to limitations of the SIF product with respect to interannual variability (Zhang et al., 2018), the evaluation of GPP against SIF is restricted to seasonal variations of photosynthesis. To reduce the impact of atmospheric transport-related uncertainties of inversion products, mean annual and seasonal variations of NEE are compared at regional scales while interannual variability is assessed at global scale. In addition, we contextualize our comparisons with FLUXCOM by providing comparisons with the previous Model Tree Ensemble (MTE) results of Jung et al., 2011 (Ju11) as well as an ensemble of process-based Global Dynamic Vegetation Model (DGVM) simulations from the TRENDY DGVM Projects (Le Quéré et al., 2018; Sitch et al., 2015). Even though FLUXCOM also produced global products of TER, these are not shown here due to a lack of an independent observational benchmark.

## 2 Data and methods

### 2.1 FLUXCOM

We used the cross-validated and trained machine learning techniques for the FLUXCOM carbon fluxes of Tramontana et al. (2016) and generated large ensembles (n = 120) of global gridded flux products for two different setups: remote sensing (RS) and remote sensing plus meteorological/climate forcing (RS+METEO) setups (Fig. 1). In the RS setup, fluxes are estimated exclusively from Moderate Resolution Imaging

Spectroradiometer (MODIS) satellite data. In RS+METEO, fluxes are estimated from mean seasonal cycles of satellite data and daily meteorological information (see Table S1). For the rationale of these setups, we refer the interested reader to Tramontana et al., 2016 and Jung et al., 2019. For the RS setup, nine machine learning methods were used to generate gridded products at an 8-daily temporal and 0.0833° spatial resolution for the 2001-2015 period. For the RS+METEO setup, three machine learning methods with five global climate forcing data sets (Table 1) yielded products with daily temporal and 0.5° spatial resolution and time periods depending on the meteorological data. The meteorological data included WATCH Forcing Data-ERA Interim (WFDEI; Weedon et al., 2014), Global Soil Wetness Project 3 forcing data (GSWP3, Kim, 2017), CRU-JRA version 1.1 (Harris, 2019), ERA5 ((C3S), 2017), and a combination of observation-based radiation from CERES (Doelling et al., 2013) and precipitation from GPCP (Huffman et al., 2001) (CERES-GPCP) resampled to 0.5°. The wide range of data sources from reanalysis to station measurements to satellite observation is intentional and is meant to bracket potential uncertainties in meteorological forcing.

For GPP and TER, we additionally considered uncertainty from flux partitioning methods by propagating two different variants, one based on night-time NEE data (Reichstein et al., 2005) and one on daytime data (Lasslop et al., 2010). Within the RS and RS+METEO setups, we followed a full factorial design of machine learning methods (9 for RS, 3 for RS+METEO), flux partitioning variants (2 for GPP and TER), and climate forcing input products (5, only for RS+METEO). Descriptions of machine learning methods, training, and validation setup are available in Tramontana et al., 2016. The methodology of generating the global products is documented in detail in the overview paper on global energy fluxes from FLUXCOM (Jung et al., 2019).

To allow for a better reuse of the large archive, we generated ensemble products of monthly values where individual ensemble members were first aggregated to monthly means (Fig. 1). The ensemble products encompass estimates of different machine learning estimates, flux partitioning variants for GPP and TER, and different climate input data for RS+METEO. For the RS+METEO setup, this was also done separately for each climate forcing data to allow modellers to compare their simulations with the FLUXCOM ensemble product driven by the same forcing. The ensemble products (hereafter referred as FLUXCOM-RS and FLUXCOM-RS+METEO) were generated as the median over ensemble members for each grid cell and month. The FLUXCOM-RS products are based on 9 ensemble members for NEE and on 18 for GPP and TER. The FLUXCOM-RS+METEO is based on 15 ensemble members for NEE and on 30 for GPP and TER.

**2.2 Process-model simulations (TRENDY)**

Dynamic Global Vegetation Models (DGVMs) represent an independent, process-based and bottom-up approach to represent the terrestrial carbon cycle and its evolution with changing environmental conditions. Here we use data from an ensemble of 16 DGVMs that were forced with the same climate (CRU-JRA v1.1), global atmospheric $CO_2$ concentration, and land-use and land cover change data (S3 simulation) over the period 1700 – 2017, following a common protocol (TRENDY-v7) (Le Quéré et al., 2018; Sitch et al., 2015). This ensemble provides fluxes at a monthly temporal resolution harmonized to a common 1° spatial resolution with simulations from: CABLE-POP, CLASS-CTEM, CLM5.0, DLEM, ISAM, JSBACH, JULES, LPJ-GUESS, LPJ, OCN, ORCHIDEE-CNP, ORCHIDEE-Trunk, SDGVM, SURFEX and VISIT. TER was calculated as the sum of heterotrophic and autotrophic respiration; NEE as heterotrophic respiration minus net primary productivity. NBP

169 from models incorporates additional fluxes as well: fire emissions (10 DGVMs), land use change (all DGVMs),
170 harvest (14 DGVMs), grazing (6 DGVMs), and any other carbon flux in/out of the ecosystem (e.g. erosion, 1
171 DGVM, VISIT). LPJ-GUESS was excluded from comparisons of NEE or NBP since monthly output on
172 heterotrophic respiration was not available.

**2.3 Independent observation-based products**

For the comparison with GPP, we used gridded monthly SIF GOME-2 (Köhler et al., 2015) retrievals from the
far-red spectral range, and for the evaluation of NEE atmospheric inversion-based estimates from Jena
CarboScope (Rödenbeck et al., 2018), CAMSv17r1 (Chevallier et al., 2005; Chevallier et al., 2019), and
CarbonTracker-EU (CTE2018, Peters et al., 2010; van der Laan-Luijkx et al., 2017). We further include
comparisons to the previous GPP and NEE upscaling products of Jung et al., 2011 (hereafter referred as Ju11).

**2.4 Comparison approach**

**2.4.1 General considerations**

All products were harmonized to a common 1° spatial resolution with monthly temporal resolution as a basis of
all comparisons shown here. Cross-consistency checks for mean annual and mean seasonal variations of GPP
and NEE are based on the three year period 2008-2010. The time period is constrained by the availability of
GOME-2 data starting in 2008 and the corresponding end year of the RS+METEO ensemble with the GSWP3
forcing ending in 2010. The NEE interannual variability was initially assessed for 2001-2010 which is the
common period of the RS and RS+METEO ensembles while comparisons for longer-time periods were also
facilitated by using meteorological forcing specific RS+METEO products that cover longer time periods (Table
1).

FLUXCOM-RS and FLUXCOM-RS+METEO products are evaluated mostly separately. We report estimates for
the respective ensemble product (see section 2.1): the spread over individual ensemble members for uncertainty
and the mean of the ensemble members; the latter can be different from the ensemble product estimate (see
Sect.2.1). Occasionally, we use the range of estimates from the union of RS and RS+METEO ensemble
members to show the full FLUXCOM uncertainty range across the two setups (labelled as "FLUXCOM" only).
For the comparison of regional or global flux values, we used flux densities rather than integrated fluxes due to
inconsistencies in land-sea masks in different products. A common mask of valid data from the intersection of
FLUXCOM, TRENDY, and Ju11 was applied to all data streams, and a land area-weighted regional or global
mean calculated. Globally integrated GPP was calculated by scaling the global mean GPP density flux with the
global non-barren land area (122.4 Mio $km^2$) derived from the MODIS land cover product (Friedl et al., 2010).
All reported $R^2$ values are squared Pearson's correlation coefficients but negative correlation signs are
maintained through by multiplying $R^2$ values by -1. We aimed at structuring the cross-consistency checks with
SIF and inversion data to minimize confounding factors and uncertainties of the independent data that may have
affected the conclusions otherwise.

## 2.4.2 Rationale of GPP-SIF comparison

As the GPP-SIF relationship is approximately linear over seasonal time scales (Zhang et al., 2016), the comparison was based on monthly values. To minimize confounding effects of canopy structure (e.g. Migliavacca et al., 2017), the comparisons were done over time when canopy structure changes relative to GPP changes are expected to be much weaker than spatial changes. The unstable orbit of the MetOp-A satellite that carries one of the GOME-2 instruments and sensor degradation effects do not permit conclusive comparisons with respect to interannual variability (Zhang et al., 2018). Therefore, we restricted the analysis to mean seasonal cycles and show 1° maps of the $R^2$ between mean monthly GPP and SIF.

There are remaining caveats and uncertainties associated with the GPP-SIF relationship (see e.g. Porcar-Castell et al., 2014 for an overview). Nevertheless, various studies have shown that SIF is currently the best proxy for photosynthesis that can be remotely-sensed directly, in particular at seasonal time scale and over regions with strong seasonal cycles. This is supported by strong empirical relationships between GPP and SIF across different satellites and retrieval methods as well as from EC data, crop inventories, and data-driven GPP methods (Frankenberg et al., 2011; Guanter et al., 2014; Joiner et al., 2018; Sun et al., 2017; Walther et al., 2016). This gives us confidence in using SIF as an independent data stream for photosynthesis to evaluate FLUXCOM products.

## 2.4.3 Rationale of comparing net carbon fluxes with atmospheric inversions

We compared atmospheric inversion-based net carbon release with FLUXCOM mean NEE at the seasonal scale over the established 11 TRANSCOM regions (see Fig.S1 for a map) as atmospheric inversions are better constrained over large spatial scales (Peylin et al., 2013). The comparison of interannual variability was conducted at global scale due to its smaller signal and larger transport uncertainties compared to the seasonal cycle. Due to various inversion uncertainties related to choices of atmospheric transport model, atmospheric station $CO_2$ data, fossil fuel information, prior constraints, driving wind fields, and inversion strategy, we used three different products: Jena CarboScope (s99oc_v4.3, Rödenbeck et al., 2018), CAMSv17r1 (Chevallier et al., 2005; Chevallier et al., 2019), and CarbonTracker-EU (CTE2018, Peters et al., 2010; van der Laan-Luijkx et al., 2017). To evaluate global NEE interannual variability patterns for periods since the late 1950s until present, we further use two long-term atmospheric inversions (CarboScope s57Xoc_v4.3, sEXTocNEET_v4.3, Rödenbeck et al., 2018) and annual $CO_2$ growth rate from the Global Carbon Budget (Le Quéré et al., 2018).

It is important to note that FLUXCOM NEE is semantically different from inversion-based net carbon exchange between land and atmosphere. The former is solely the difference between gross fluxes (i.e., NEE = TER - GPP) while the latter integrates all vertical movement of $CO_2$ including, for example, fire emissions, evasion from inland waters, respired harvests, or volatile organic compounds (Kirschbaum et al., 2019; Zscheischler et al., 2017). Simulations from TRENDY models report both, NEE and net biome productivity (NBP) which is conceptually close but not identical to what atmospheric inversions provide. To assess whether conclusions are affected by the different NEE vs NBP definitions we a) provide NEE and NBP estimates from TRENDY models, b) we include comparisons where inversions were corrected for fire emissions (from CarbonTracker-EU) to yield estimates closer to NEE, and c) discuss whether discrepancies with FLUXCOM can originate from the omission of secondary carbon loss pathways given in the literature.

**3 Results and discussion**

**3.1 Gross primary productivity**

**3.1.1 Mean annual gross primary productivity**

Overall, our results suggest a high degree of cross-product (and, for FLUXCOM, also within-product) consistency of global mean GPP patterns (Fig. 2). In fact, global patterns of mean GPP are consistent across both FLUXCOM ensembles ($R^2$=0.97) as well as for Ju11 and TRENDY ensemble mean ($R^2$>0.94), despite sizeable regional differences. The slope of the pair-wise 1:1 regressions among the different mean GPP data sets varies within ~10%. FLUXCOM-RS shows about 10-20% lower GPP than FLUXCOM-RS+METEO in the highly productive tropics and some subtropical regions. Both FLUXCOM setups estimate larger GPP than Ju11 and TRENDY in some semi-arid regions and about 5-15% lower GPP in some extratropical areas. Despite a sizeable total range of mean GPP from all 48 FLUXCOM members, the majority of TRENDY models (at least 9 out of 16) fall outside the FLUXCOM range for about 70% of the land surface (Fig. 3).

The mean global GPP of FLUXCOM-RS (111 PgC yr$^{-1}$) is about 10% lower than RS+METEO (120 PgC yr$^{-1}$, Fig. 4), which is largely driven by differences in the tropics (Fig. 2). The cross-validation analysis indicated an underestimation of FLUXCOM-RS GPP in the tropics (Tramontana et al., 2016), which was confirmed by a grid cell-to-site data comparison for the FLUXNET 2015 data (which were not used for machine learning training here) (Joiner et al., 2018). The reasons for the on-average lower GPP of RS compared to RS+METEO require further investigation. It is unlikely that the smaller RS GPP values are because this setup is exclusively based on remote sensing, as global latent heat from RS was larger than Ju11 (Jung et al., 2019). It seems to be rather related to the specifically different predictor sets between RS and RS+METEO. This indicates that future FLUXCOM efforts should expand the ensemble with respect to predictor set diversity to better account for this source of uncertainty in upscaling. Focussing on FLUXCOM-RS+METEO, its ensemble spread (108-130 PgC yr$^{-1}$) is much smaller than the TRENDY-based global GPPs (83-172 PgC yr$^{-1}$), and is primarily due to differences among machine learning methods rather than meteorological forcing data (Fig.S2).

Our results imply that the present FLUXNET upscaling approach does not agree with larger GPP values of 150-175 PgC yr$^{-1}$ derived from an isotope-based study (Welp et al., 2011). It is possible that the FLUXNET upscaling approach underestimates GPP of highly managed and fertilized crops (Guanter et al., 2014) but their effects on global GPP biases seem small (Joiner et al., 2018). At FLUXNET sites night-time $CO_2$ advection and storage could cause underestimation of night-time $CO_2$ fluxes (Aubinet et al., 2012; McHugh et al., 2017; van Gorsel et al., 2009) and thus underestimate GPP using the night-time NEE flux partitioning method. On the contrary, it has been suggested that FLUXNET GPP estimated from the night-time partitioning method (Reichstein et al., 2005) is overestimated as it ignores the effects of light inhibition of leaf respiration (Keenan et al., 2019; Wehr et al., 2016) by on average 7% across FLUXNET sites (Keenan et al., 2019). But it should be noted that this value may not be globally representative due to sizeable variations between ecosystems and with leaf area. Further, we only find a small difference of mean global GPP of <2 PgC for day-time (Lasslop et al., 2010) and night-time (Reichstein et al., 2005) NEE partitioning. This suggests that neither $CO_2$ advection nor the light inhibition of leaf respiration appear to generate sizeable biases of global GPP in FLUXCOM—a tendency likely encouraged by the relatively strict quality control on the EC fluxes data (Tramontana et al., 2016). Furthermore, a

comparison of EC-based GPP with biometric GPP estimates across 18 globally distributed sites showed good agreement and no significant bias (Campioli et al., 2016). A recent study using Carbonyl Sulfide (COS)-based partitioning for four contrasting European sites also showed good agreement with standard EC-based GPP where systematic differences for mean GPP were < 5% (Spielmann et al., 2019). Therefore, we currently have no strong indication that systematic biases of FLUXNET GPP propagate to global FLUXCOM GPP. Nevertheless, we need to acknowledge that global GPP is largely driven by the productivity in the tropics where flux towers are scarce and may be particularly uncertain due to challenging logistic and micrometeorological conditions (Fu et al., 2018).

Various remote sensing-based light use efficiency approaches, calibrated with flux tower data, yielded global GPP estimates of 109 (Zhao et al., 2005), 111±21 (Yuan et al., 2010), 108-119 (Yu et al., 2018),122±25 (Jiang and Ryu, 2016), 132±22 (Chen et al., 2012), and 140 PgC yr$^{-1}$ (Joiner et al., 2018). A simple calibration of only near-infrared reflectance (NIRv) to EC data suggested a global GPP of 131-163 PgC yr$^{-1}$ (Badgley et al., 2019). Studies that assimilated atmospheric $CO_2$ concentration data into process model simulations yielded slightly higher values of 148 (Anav et al., 2015) and 146±19 PgC yr$^{-1}$ (Koffi et al., 2012) with the latter study unable to distinguish their best estimate from a global GPP of 117 PgC yr$^{-1}$ because the atmospheric $CO_2$ alone cannot constrain magnitudes of gross fluxes well. Assimilating SIF into process-models yielded 137±6 (Norton et al., 2019) and 166±10 PgC yr$^{-1}$ (MacBean et al., 2018). More recent isotope studies derived global GPP as 120±30 PgC yr$^{-1}$ (Liang et al., 2017), and global NPP of ~60 PgC yr$^{-1}$ (Hellevang and Aagaard, 2015) which implies global GPP of 109-150 PgC yr$^{-1}$ considering a range of NPP:GPP ratios of 0.4-0.55. In conclusion, global FLUXCOM GPP estimates are within the currently most plausible 110-150 PgC yr$^{-1}$ range.

**3.1.2 Seasonal cycles of gross primary productivity**

Cross-consistency analysis of mean monthly GPP seasonal cycles from FLUXCOM with SIF from GOME-2 (Köhler et al., 2015) shows widespread and strong agreement for both FLUXCOM setups (Fig. 5), except for the inner tropics where seasonality is weak and SIF retrievals might be affected by the South Atlantic Magnetic Anomaly (Köhler et al., 2015). FLUXCOM-RS tends to show better agreement with SIF than FLUXCOM-RS+METEO in agricultural regions of Southeast Asia, maybe because only the mean seasonal cycles of remotely sensed land surface properties were used in the latter. Conversely, FLUXCOM RS+METEO shows on average better consistency with SIF in some semi-arid regions, e.g., Australia. However, maps of the maximum $R^2$ with SIF for RS and RS+METEO respectively have similar patterns with good agreement of both products in Australia, and even in the tropics (Fig.S4). This suggests that the inclusion of some machine learning methods somewhat negatively impacts the ensemble, especially for RS which shows larger spread (see Fig.S4 for mean $R^2$ of the RS ensemble members). With SIF, both FLUXCOM setups show similar consistency as Ju11. The consistency of FLUXCOM with SIF is much better than with TRENDY models, in particular in tropical and subtropical regions. This implies that, despite sporadic spatial coverage of FLUXNET sites and previously identified incomplete capturing of water stress (Bodesheim et al., 2018; Tramontana et al., 2016), FLUXCOM still has a large potential to inform and constrain process-based model simulations of seasonal variations of photosynthesis in moisture-limited regions.

**3.2 Net ecosystem exchange**

**3.2.1 Mean annual net ecosystem exchange**

In most TRANSCOM regions, FLUXCOM shows a stronger mean annual net carbon uptake than indicated by atmospheric inversions with a particularly large systematic difference in the tropics (Fig. 6). This pattern of a large tropical carbon sink in FLUXCOM is qualitatively consistent among the different FLUXCOM setups and ensemble members, as well as with previous estimates from Ju11. To date, this is a systematic feature of the current data-driven approach of upscaling EC measurements with machine learning.

Multiple independent approaches indeed imply a sizeable carbon sink in intact tropical forests (Arneth et al., 2017; Gaubert et al., 2019; Pan et al., 2011), which appears to be largely or entirely offset by carbon loss pathways in the tropical region such as fire, land-use change emissions, and evasion from inland waters. These $CO_2$ sources are not sampled by EC measurements from FLUXNET, and are, therefore, not represented in FLUXCOM. However, the missing fluxes only resolve up to roughly half of the gap (Zscheischler et al., 2017). The comparatively small differences between net carbon release estimates by inversions and those where fire emissions were corrected for, as well as the small differences between NEE and –NBP from TRENDY further suggest that these secondary carbon loss fluxes do not drive the large discrepancy between FLUXCOM and inversion-based mean net carbon exchange. Nevertheless, substantial uncertainty remains in the magnitude of these secondary carbon fluxes and their incomplete accounting in TRENDY models and inversions (Kirschbaum et al., 2019; Zscheischler et al., 2017).

Issues with the current FLUXCOM approach certainly contribute, likely dominate, the discrepancy between atmospheric top-down and FLUXCOM mean NEE. Potential factors that could contribute to this are (1) a FLUXNET sampling bias (see also Sect. 4.1.2) towards ecosystems with a large carbon sink, particularly in the tropics (Saleska et al., 2003); combined with (2) missing predictor variables related to disturbance and site-history (Amiro et al., 2010; Besnard et al., 2018, see also Sect. 4.2.1), or (3) biases of eddy covariance NEE measurements, e.g. due to night-time advection of $CO_2$ (Hayek et al., 2018; van Gorsel et al., 2008), especially under tall tropical forest canopies (Hutyra et al., 2008, Fu et al., 2018). Fu et al. (2018) studied 63 site-years of EC data from 13 tropical forest sites and report a mean between-site NEE of -567 gC m$^{-2}$ yr$^{-1}$ showing that the large tropical sink in FLUXCOM is inherited from FLUXNET data. The authors pointed out that for about half of the sites where measurements of $CO_2$ concentration along the vertical profile were available and the storage was considered in the NEE processing, the carbon sink was less than half (-340 gC m$^{-2}$ yr$^{-1}$) compared to those without storage correction (-832 gC m$^{-2}$ yr$^{-1}$). However, the small sample size together with the large between-site standard deviation of mean NEE (459 gC m$^{-2}$ yr$^{-1}$) not only makes robust conclusions difficult, but also indicates potentially large diversity between tropical ecosystems. Clearly, more tropical EC sites are needed along with a better accounting of systematic errors in EC-based NEE measurements to resolve this issue.

**3.2.2 Seasonal cycles of net ecosystem exchange**

We find a good consistency between FLUXCOM and inversions with respect to amplitude and shape of the seasonal cycles of NEE in many TRANSCOM regions, especially over the North American Boreal, North American Temperate, and Europe regions with $R^2$ values > 0.92 (Fig. 7). As with mean annual NEE, the

seasonal cycle mismatch relative to inversions may be linked to carbon loss fluxes not accounted for in FLUXCOM, such as fire emissions that are seasonally relevant in tropical and subtropical regions. However, adjusting inversion-based NBP towards NEE by correcting for fire emissions does not improve the correspondence with FLUXCOM in tropical and subtropical regions (Fig.S5). In tropical regions, the weak seasonality paired with comparatively large spread among inversions does not allow for robust conclusions. Overall, the seasonal variations of FLUXCOM NEE show potential to constrain the large uncertainty in TRENDY models, and potentially even atmospheric inversions at the regional scale, especially considering that their uncertainty range across only three products is still significant.

### 3.2.3 Interannual variability of net ecosystem exchange

Spatial patterns of the magnitude of the interannual variability (IAV) of land carbon sink for the period 2001-2010 share some common features among atmospheric inversions, FLUXCOM-RS, FLUXCOM-RS+METEO and TRENDY. For example, all products identify the hotspots in southeast Asia, southern North America, and also in the Siberian tundra (Fig. 8). Overall, there are still differences in the spatial patterns of IAV magnitude among and within different data-streams.

All EC data-driven methods, in particular FLUXCOM-RS+METEO, underestimate magnitude of IAV compared to inversions (Fig. 8). The reasons for the underestimation of IAV magnitude by FLUXCOM are not fully clear. Within FLUXCOM, the smaller IAV magnitude of RS+METEO NEE compared to that of RS is linked to the use of only mean seasonal cycles of RS-based land surface properties in RS+METEO setup. The IAV of carbon loss fluxes that are not captured by FLUXCOM, such as through fire, are currently thought to be comparatively small at the global scale and appear minor here (see Fig.S6). Machine learning methods already underestimate the IAV at the site level (Marcolla et al., 2017; Tramontana et al., 2016). The low bias in FLUXCOM IAV is a direct consequence of the comparatively small explained variance for NEE anomalies. Thus, improving the predictability of NEE IAV at site level has potential to also correct the magnitude of globally integrated IAV variance.

Despite the tendency of FLUXCOM products to underestimate IAV magnitude, FLUXCOM-RS+METEO reproduces year-to-year variations of globally integrated annual land carbon exchange anomalies derived from atmospheric inversions for 2001-2010 ($R^2$=0.87). It shows better consistency than TRENDY with one of the long-term inversions (Fig.S7). Further examination of this ensemble reveals that the choice of machine learning method, rather than meteorological forcing data, has a larger influence on IAV of global NEE (Fig.S8). Here, the Random Forests method performed less well compared to the other two methods. Interestingly, training Random Forests with an almost identical predictor set but at half-hourly temporal scale rather than at daily scale (Bodesheim et al., 2018) substantially improved the $R^2$ (from 0.31 to 0.60, S8). This indicates that machine learning methods can benefit from higher temporal variability provided by millions of high-frequency NEE measurements, especially for signals such as IAV that are small and difficult to extract. In addition, underlying functional relationships can be better extracted from high-frequency data as the predictor space is better covered, allowing for improved discrimination of drivers that have stronger covariation on longer time-scales.

To better understand the qualitatively different global NEE IAV patterns between RS and RS+METEO setups,
we infer which NEE IAV signals are consistent or lacking among FLUXCOM setups and TRENDY models by
assessing correlation patterns (Fig. 9). We find the strongest consistencies of NEE IAV between FLUXCOM-RS
and FLUXCOM-RS+METEO in many semi-arid regions, and almost no consistency otherwise. This suggests
that the main discrepancies of globally integrated NEE IAV between FLUXCOM-RS and FLUXCOM-
RS+METEO are likely not due to differences in their capabilities of reflecting water stress effects. It has been
shown that despite the local dominance, water-related NEE anomalies largely cancel spatially in RS+METEO
and TRENDY resulting in the dominance of temperature-related NEE anomalies in globally integrated land sink
IAV (Jung et al., 2017, but see Humphrey et al., 2018 for a different perspective). Studies on effects of water
availability on spatial GPP anomalies using the RS data yielded highly plausible patterns that were consistent
with independent data (Flach et al., 2018; Orth et al., 2019; Walther et al., 2019). Also the comparison of
FLUXCOM-RS GPP monthly anomalies with the independent FLUXNET2015 data set showed unexpected
large consistency when anomalies were scaled by the site-specific observational range (Joiner et al., 2018).
When delineating the regions with larger agreement between RS+METEO and TRENDY than that between RS
and TRENDY, we can infer that FLUXCOM-RS seems to miss important NEE anomaly features in the tropics.
This is likely due to (1) a combination of sparse satellite data availability, cloud contamination, and geometrical
illumination effects in the tropics or (2) that the processes governing NEE IAV in the tropics cannot be captured
by satellite-based predictors alone in RS (even under ideal observational conditions) but require additional
meteorological variables such as temperature that is included in the RS+METEO setup. Some support for the
latter point comes from Byrne et al., 2019 who found strong correlations of anomalies from GOSAT inversions
with NEE from RS+METEO and soil temperatures in the tropics but not with SIF and a drought indicator,
suggesting that temperature impacts respiration more than photosynthesis in the tropics.

Overall there are large discrepancies among FLUXCOM and TRENDY as well as amongst TRENDY models
with respect to local NEE IAV. This reflects our limited understanding and capabilities to model year-to-year
variations of local ecosystem carbon exchange. Both data-driven and process-based approaches also showed
poor performance with respect to NEE IAV in FLUXNET sites (Tramontana et al., 2016, Morales et al., 2005).
However, both approaches yield good correspondence of globally integrated NEE with atmospherically-derived
interannual land sink variations. This correspondence is due to two reasons: first, the spatial compensation of
locally important processes that are not well captured by the models; and second, models capture better the
temperature-related signals that gain relevance at larger spatial scales (Jung et al., 2017). Whether the large
uncertainty of modelling NEE IAV at ecosystem level is due to misspecified parameterizations, missing
predictors, inaccurate forcing data and/or absent processes remains a research priority. Our understanding and
ability to model NEE IAV bottom-up would greatly benefit from atmospheric inversions that could localize NEE
robustly. Exploiting the massive space-based column $CO_2$ data in the future will hopefully facilitate the
improvements on this aspect. Despite large uncertainties and apparent knowledge gaps in NEE IAV from both an
observational and modelling perspective, there are promising indications of improved capability to track IAV
patterns with FLUXCOM such as the good correspondence of RS+METEO with inversions at global scale, and
independent verifications of GPP IAV of RS at least outside the wet tropics (Flach et al., 2018; Joiner et al.,
2018; Orth et al., 2019; Walther et al., 2019).

## 4 Methodological limitations and potential ways forward

Machine learning methods can learn arbitrarily complex functions and provide a nearly perfect model of a phenomenon if they are fed with the right data and trained thoroughly. Thus the quality, quantity, and completeness of the input data determine the quality of the output. In the following, we discuss the relevance of limitations associated with data from the FLUXNET network, and of the limited capabilities of representing all relevant factors by observable predictor variables. We also outline potential strategies for improvements, both overall and with respect to machine learning approaches specifically. The continued and rapid development of machine learning notwithstanding, we believe that the FLUXCOM approach is at present more limited by available "information" rather than by available machine learning methods.

### 4.1 FLUXNET observations

### 4.1.1 Potential observation errors

The comparatively large random errors of high-frequency EC measurements diminish quickly when aggregated to daily or 8-daily averages used here. Furthermore, training on half-hourly EC data (Bodesheim et al., 2018) helps machine learning methods extract patterns from noisy data. In general, poor signal-to-noise ratios can be counteracted by larger sample size. More problematic than random errors are potential systematic errors of EC measurements since those would propagate to the derived global carbon flux products. Even though there have been large efforts by the community to characterize and to correct for systematic errors, such as those due to low turbulence and $CO_2$ advection (e.g. Aubinet et al., 2005; Aubinet et al., 2012; Papale et al., 2006), uncertainties remain on the relevance and magnitude of those errors in the processed FLUXNET data. Differences due to instrumentation and maintenance pose another potential source of uncertainty. Additionally, the energy balance closure gap at FLUXNET sites is still not resolved (Stoy et al., 2013), while it remains unclear to what extent this is relevant for $CO_2$ fluxes (Leuning et al., 2012). Systematic errors in GPP and TER derived from the flux partitioning method of NEE based on night-time data (Reichstein et al., 2005) may arise due to the neglected effect of inhibited photorespiration during daytime (Keenan et al., 2019; Wehr et al., 2016). Nevertheless, all these issues together seem to be relatively small compared to the predominant patterns of variability in EC data, e.g., seasonal variations, that are very consistent across FLUXCOM and independent observation-based data streams shown here. The relatively strict quality controls on the flux training data (Tramontana et al., 2016) may have been instrumental here. The trade-off between data quality and training data volume was not explicitly studied in FLUXCOM, and related experimental setups would be desirable to gauge the robustness of the global products shown here. Even small systematic errors in EC data could degrade important signals such as interannual variability, trends, annual sums of NEE, or subtle differences between sites related to functional properties (e.g., radiation use efficiency). Systematic errors that would be prevalent across the network would result in systematic biases of derived global fluxes. For global GPP and energy fluxes (Jung et al., 2019), the values obtained from FLUXCOM are generally consistent with current knowledge but our ability to independently quantify such fluxes is also limited.

### 4.1.2 Potential representation issues

Ideally, a measurement network samples all relevant gradients of the driving factors and magnitudes of the predicted quantities. There are several potential issues with the current sampling by FLUXNET sites. With

respect to relevance for net carbon exchange, there are carbon loss pathways that FLUXNET does not capture such as fire emissions, $CO_2$ evasion from inland waters, and lateral exports due to harvest or erosion that are respired elsewhere (Kirschbaum et al., 2019). The effects of strongly enhanced respiration in the years after large disturbances (Amiro et al., 2010) are challenging to capture due to stochastic and destructive nature of disturbances.

To meet the assumptions of EC method, FLUXNET stations are confined to reasonably flat terrain. Topographic effects on ecosystem fluxes are primarily due to their influence on environmental drivers, i.e., the predictor variables. Thus, the extrapolation to hillslopes should be reasonable if the topographic effects are accounted for in the gridded predictor variables. This might be challenging especially for remote sensing products due to necessary but complicated corrections of illumination conditions. The uncertainties of these topographic factors might become particularly relevant and should be studied for prediction of fluxes at a higher spatial resolution. For the current FLUXCOM products with rather coarse spatial resolution, we expect that topographic effects are reflected in the predictor variables and the remaining subpixel heterogeneity largely cancel out.

Perhaps the most fundamental and frequent critique of the FLUXNET upscaling approach is related to the spatially clumped geographic distribution of EC sites in North America, Europe, Japan, and now Australia with only sparsely distributed towers elsewhere (Schimel et al., 2015). However, what matters eventually for machine learning methods is how well the predictor space, rather than geographic space, is sampled. To assess this, we developed an extrapolation index (EI) that estimates the expected additional relative error of a flux prediction due to a large distance to the nearest training data in the predictor space (S2). We applied this method for GPP and FLUXCOM-RS training data as an example, and found that the conditions that are least well represented by FLUXNET are associated to primarily extremely cold and dry regions (Figure 10). Surprisingly, the humid tropics are well represented in the predictor space suggesting that the environmental conditions represented by the predictor set are well sampled by the data from FLUXNET sites. The extremely cold and dry conditions that seem to constitute the biggest extrapolation issues are typically associated with small GPP fluxes and thus also small prediction errors. To account for that, we spatialized the expected GPP error of the RS ensemble (Figure 10, see S2 for details), which largely scales with GPP magnitude but also shows patterns of larger expected errors in semi-arid regions than that expected from flux magnitude alone. The multiplication of the expected GPP error with the extrapolation index provides the extrapolation severity index (ESI) that shows where poor FLUXNET sampling likely increases the absolute prediction error strongly. According to these results, sub-tropical semi-arid regions, in particular India, appear as most affected, suggesting that GPP upscaling from FLUXNET would benefit most strongly from improved data availability for towers representing these conditions. Despite these limitations of data, we found excellent consistency of FLUXCOM GPP seasonal cycles with SIF over these regions, which was in fact much better than the consistency between TRENDY models and SIF. This suggests that while more towers in semi-arid regions will help reduce uncertainty in future upscaling efforts, FLUXCOM can already provide useful information for constraining the models in these regions. It also shows that the bias in geographic representation of FLUXNET sites is not as critical as anticipated due to the flexibility and adaptiveness of machine learning methods. The sampled environmental conditions (predictor space) should cover the conditions of the global application domain rather than being representative of it. The

larger issue of the FLUXNET representation bias is associated with drawing conclusions from the site-level cross-validation because the evaluation metrics are easily biased towards certain regions and ecosystems.

The methodology used here to assess the extrapolation problem quantitatively has several limitations. For example, potential differences in EC data quality were not accounted for. Perhaps, the largest but unavoidable limitation is the reliance on the predictor set and the assumption that it captures all relevant gradients. In a sense, the methodology can only uncover "known unknowns". If an important predictor is missing, the method would, of course, not see any extrapolation penalty with respect to the missing factor. Somewhat ironically, we may need more towers in the first place to identify further relevant predictors in an objective way to, say, better capture the diversity in the tropics (Fu et al., 2018) or in agricultural systems (Guanter et al., 2014) where we anticipate that the current sampling is limiting the FLUXCOM approach.

**4.2 Driving factors and predictors**

Assuming infinite sample size, perfect quality and coverage, the success of machine learning methods depends entirely on the completeness of the predictor set for the target variable, given an adequate training. The predictor set for FLUXNET upscaling is practically constrained by 1) the availability of consistent observations at site level across all sites, and for most of their temporal coverage at a spatial resolution sufficiently close to the flux tower footprint; and 2) the availability of corresponding global grids at an adequate spatial and temporal resolution and temporal coverage. This explains the predictor space of remotely sensed land products from MODIS along with tower-measured meteorology chosen in FLUXCOM. While the general success of the FLUXCOM approach suggests that the predictor sets contain sufficient information for predicting the variability of carbon fluxes, it is also obvious that some factors are not well accounted for.

**4.2.1 Site-history**

It has been argued previously (Besnard et al., 2018; Jung et al., 2011; Tramontana et al., 2016) that the current limitations of unrealistic mean NEE patterns from FLUXNET upscaling is also due to missing predictor variables that describe site history effects such as forest age or time since disturbance. These factors have been shown to influence IAV (Musavi et al., 2017; Tamrakar et al., 2018) and to drive mean NEE patterns in synthesis studies (e.g. Amiro et al., 2010). Including forest age in a simple empirical model helped predicting between site variations of mean NEE across FLUXNET sites (Besnard et al., 2018). Counterintuitively, including forest age in training a machine learning method on monthly NEE did not improve the predictability of mean site NEE (Besnard et al., 2019), albeit possibly due to data or methodological limitations. We find the largest discrepancies of mean FLUXCOM NEE with atmospheric inversions in the tropics, where site history plays a substantial role in NEE magnitude (Pugh et al., 2019), but the concept of forest age is hardly applicable due to the generally uneven aged nature of stands, and reliable estimates of gridded age, e.g., from forest inventories are not available. Efforts to incorporate the information from long-term LANDSAT time series to capture site history effects did not reveal an improvement in the predictions of mean NEE, but it remains unclear if this was due to limited information content in these time series or due to methodological issues (Besnard et al., 2019). Thus, this issue remains a significant scientific challenge. Potentially, the availability and application of high-resolution biomass and vegetation optical depth estimates from radar remote sensing along with a carefully

collected ancillary data on biomass, basal area, tree diameter and tree age distributions at ICOS and NEON sites
may help in the future.

**4.2.2 Management**

We are presumably lacking important information on anthropogenic management effects, in particular for crops
(Guanter et al., 2014) but also for forests. This is primarily due to a lack of information on, e.g., crop type,
fertilizer application, irrigation, harvest or thinning at FLUXNET sites, but also due to the still-limited number of
crop sites to provide sufficient information on relevant predictors therein. Accounting for the management
effects in the FLUXCOM approach either by explicit management information or implicitly by adequate remote
sensing data may also help improve the predictions of IAV of local-scale carbon fluxes, in particular with cross-
validation since most FLUXNET sites are subject to some degree of management.

**4.2.3 $CO_2$ fertilisation**

FLUXCOM lacks any explicit treatment of the effects of $CO_2$ fertilization causing carbon flux trends to be
unrealistic (Fig.S11). This is a challenging problem due to a comparatively small size of $[CO_2]$ effect. This, in
turn, makes it particularly vulnerable to distortions through measurement uncertainties, and, on an annual scale,
largely indistinguishable from any other factor that varies with time. Potentially, in the future, the availability of
longer time series along with high-quality near surface atmospheric $CO_2$ data at high spatial and temporal
resolution at the tower scale could allow for extracting a $CO_2$ fertilization effect by exploiting diurnal, seasonal,
and spatial $CO_2$ gradients in addition to the long-term trend.

**4.2.4 Water stress**

Site-level cross-validation analysis (Bodesheim et al., 2018; Tramontana et al., 2016) indicated that soil moisture
effects on carbon fluxes are not always well captured. In RS+METEO, moisture effects are explicitly addressed
by a simple meteorology driven water availability index. The RS setup relies entirely on indirect information
encoded in remotely sensed surface properties such as vegetation indices and land surface temperatures. The
comparison of FLUXCOM GPP seasonal cycles with SIF yielded excellent agreement, also in water limited
systems, and studies on drought effects using the GPP RS product (Flach et al., 2018; Orth et al., 2019; Walther
et al., 2019) found plausible patterns that were consistent with independent data on large scales. Nevertheless,
we should strive further to improve water stress effects in the upscaling approach given its significance. Better or
explicit predictor variables on soil moisture may help. Unfortunately, current soil moisture products from remote
sensing are only representative of the top few centimeters and are at comparatively coarse spatial resolution
limiting their applicability in reflecting spatial heterogeneities of soil moisture. Perhaps, the larger issue is
diverse ecosystem specific responses to soil moisture variations due to different ecosystem compositions, rooting
patterns, plant hydraulics, stomata and other physiological traits. Thus, exploring remotely sensed products that
reflect additional or complementary information on water stress effects, such as diurnal cycles of land surface
temperature from geostationary satellites, is a potential way forward.

**4.2.4 Product properties**

The success of incorporating novel informative data of site properties in the FLUXCOM approach is always
contingent on the quality of the corresponding global gridded products. Systematic differences between a
predictor variable used for training at the site-level and global forcing data, as well as any potential artefacts due
to retrieval issues or merging different data records spatially or temporally propagate to global flux products.
Future improvements of the FLUXCOM approach will thus require progress in other research fields with
emphasis on the processing, correction, and harmonization of Earth observation products. Especially for
remotely sensed data, strategies to bridge scales of satellite pixels, overpass times, and repeat cycles to
continuous measurements of flux footprints are needed. In addition, making use of novel data in the FLUXCOM
framework requires the concurrent development of new methodological strategies to cope with the small
temporal overlap of the FLUXNET data history. More generally, the quality and quantity of Earth observation
data has been increasing rapidly, bringing challenges and opportunities for upscaling.
**4.3 Machine learning**
**4.3.1 Exploiting temporal data structures**
The machine learning methods employed in FLUXCOM are classic ones, while novel approaches could bring
further improvements. One conceptual limitation of all machine learning methods used in FLUXCOM is that
they assume independent and identically distributed (i.i.d.) variables, and thus do not respect or exploit temporal
structures in the training data. This problem can be remedied by using other machine learning methods based on
convolutions. For example, recurrent neural networks (RNNs) were designed for time-series and can account for
dynamics such as ecosystem lag and memory effects on carbon flux variability. Conceptually, lag and memory
effects emerge due to the effect of unobserved ecosystem state variables. RNNs can potentially counteract the
lack of a relevant state variable in the predictor set if the state variable's instantaneous effect is encoded in the
temporal history of other predictor variables (e.g., current soil moisture as a function of previous weather). While
exploiting the temporal information of predictors using an RNN improved predictions of monthly carbon fluxes
in terms of the seasonal cycle and thereby also across-site variability, predictions of interannual variability were
not improved as compared to exploiting only time-instantaneous effects based on site-level cross-validation
(Besnard et al., 2019). Further exploration of the machine learning methods that exploit the temporal structure of
predictors has a potential to improve FLUXCOM upscaling.
**4.3.2 Promising strategies**
Deep learning techniques, in general, and convolutional neural networks (CNNs), in particular, have proven to
be very powerful especially for image processing and recognition tasks (LeCun et al., 2015). Their conceptual
strength lies in the automated extraction of features, in particular those related to spatial structures that render the
design and implementation of hand-crafted predictor variables unnecessary. Whether simply employing CNNs
for upscaling brings similar improvements over traditional machine learning techniques as in other domains is
questionable. This is because the number and spatial distribution of FLUXNET towers seems insufficient to
exploit the power of CNNs to extract relevant features of spatial structure. However, combining CNNs with
transfer learning approaches seems very promising from a conceptual perspective. The principle of transfer
learning is to learn relevant features from a more densely observed proxy variable of the actual target and use the
feature representation for learning the target (Pan and Yang, 2010). The learning of the proxy variable can be
done either prior to or simultaneously with the actual target such that information from much larger sample of
the proxy can be transferred to the sparsely observed target variable. This approach could be applicable to the
upscaling of FLUXNET GPP by using remotely sensed SIF as a proxy and thereby alleviate issues related to
small sample size (e.g., extrapolation) but also aid the extraction of small but relevant signals (e.g., IAV). Spatial
structures in high-resolution SIF data may further encode effects of management or topographically controlled
soil moisture variations that could be exploited with CNNs and improve predictions.

Hybrid approaches, i.e. the integration of machine learning method with process understanding and physical
constraints, are another promising avenue. This allows for different strategies and levels of complexity are
possible (Reichstein et al., 2019), and could also greatly help in regularizing machine learning predictions to be
sensible under extrapolation conditions. In the context of FLUXCOM, for, say, constraining the anticipated weak
signal of $CO_2$ fertilization in observations within theoretically derived bounds, would allow this relevant yet
observationally poorly constrained dynamic to be incorporated. If the hybrid approach features the
conceptualization of fluxes and pools as in process models, it would also allow for constraints by multiple
complementary data streams simultaneously.

An important aspect to improve in the future is also the quantification of uncertainty in the predictions, including
the propagation of observational uncertainties. Gaussian processes are now computationally tractable for big data
problems while providing probabilistic confidence intervals and allowing for uncertainty propagation (Camps-
Valls et al., 2016; Wang et al., 2019). Combining Gaussian Processes with deep neural nets (You et al., 2017) or
designing deep Gaussian process models (Damianou and Lawrence, 2013) are powerful new machine learning
tools with the potential to improve FLUXCOM.
**Conclusions**
The FLUXCOM initiative generated a large ensemble of global carbon flux products for two defined setups that
differ in the set of predictor variables and spatial-temporal resolution. The ensemble is comprised of 120
products using up to 9 machine learning algorithms, two flux-partitioning variants for GPP and TER, and 5
meteorological forcing data sets. The large and systematically generated ensemble allows for assessing and
studying uncertainties of the fluxes as well as the approaches used in FLUXCOM. We assessed FLUXCOM
GPP and NEE patterns against remotely sensed sun-induced fluorescence (SIF), atmospheric inversions and
process model simulations from the TRENDY initiative.

We found strong consistency of FLUXCOM with SIF and atmospheric inversions with respect to seasonal
variations, highlighting FLUXCOM's suitability to evaluate and constrain seasonal cycles for processed-based
and top-down approaches. The global GPP from RS+METEO was 120±7 PgC yr$^{-1}$ (mean±1 s.d.), while the
global GPP from RS (111±3 PgC yr$^{-1}$) is lower likely due to underestimation in the tropics. FLUXCOM shows a
consistently large carbon sink in the tropics that can, at present, not be reconciled with our knowledge derived
from atmospheric $CO_2$ constraints; possibly implying an underestimation of carbon loss and/or missing carbon
loss pathways by FLUXNET observations. Patterns of year-to-year variations of the global land carbon sink
from FLUXCOM-RS+METEO show good consistency with atmospheric inversions, while magnitudes of
interannual variability are underestimated in the data-driven approaches. As FLUXCOM lacks the effect of $CO_2$
fertilization, trends are not realistic and should only be used for assessing the exclusive effects of climate
changes on carbon fluxes.

Moving forward, increasing the size of the FLUXNET network, improving its quality, standardization and
coverage will both improve quality and reduce uncertainties in the upscaling approach. This holds especially
with respect to signals that are important but relatively small and difficult to extract such as interannual
variability or trends. Increasing the number of tropical sites alone would also help constrain global flux
magnitudes, and, in particular, would help resolve the large tropical carbon sink shown by FLUXCOM but
missing in atmospheric inversions. Based on the number of registered FLUXNET sites alone, an approximate
five-fold increase in the number of sites with available data seems feasible in theory; if all respective researchers
would contribute their flux data to the global community effort. This indicates that any efforts to improve eddy
covariance data, sharing, harmonization and processing are crucial.

Beyond extending the data frame, the current FLUXCOM intercomparison suggests that the next phase of
methodological developments should be to move away from predetermined setups and instead towards a set of
dedicated experiments that explore novel strategies of data integration with machine learning method (e.g., deep,
transfer, and hybrid approaches) and, more importantly, the diversity in the potential predictor space from Earth
Observation data. Within FLUXCOM, we find the largest differences between RS and RS+METEO setups
which primarily differ in the set of input predictor variables. Thus, the current approach of upscaling FLUXNET
measurements seems more information rather than algorithm limited.

Overall, the success of FLUXCOM approach depends on the interplay of many different factors. Monitoring our
progress will be essential but challenging, and must combine site-level cross-validation, cross-consistency
checks with global independent data-streams, novel and dedicated experiments as well as tailored validations of
methods with artificial data similar to Observation System Simulation Experiments. Despite the many
challenges, integrating eddy covariance ecosystem scale fluxes, Earth Observation data and machine learning
method has already proven valuable in many respects despite being a comparatively new field. An exciting and
challenging future lays ahead; that the contribution of experts in different fields combined with open and real
time data sharing could lead to a unique semi-operational carbon monitoring system. This in turn provides a
promising perspective to unify and synergistically exploit data-driven biospheric bottom-up and atmospheric
top-down approaches.
**Data availability**
Monthly carbon flux data of all ensemble members as well as the ensemble estimates from the FLUXCOM
initiative (http://www.fluxcom.org) are freely available (CC4.0 BY licence) from the data portal of Max Planck
Institute for Biogeochemistry (https://www.bgc-jena.mpg.de/geodb/projects/Home.php) after registration.
Choose 'FluxCom' in the dropdown menu of the database and select FileID 260. The users will be provided with
an access to an ftp server. The ftp directory is structured in a consistent way and stores files with consistent
naming convention in netcdf-4 format (see S3 for details).. Products with daily or 8-daily temporal resolution or
customized ensemble estimates are available on request to Martin Jung (mjung@bgc-jena.mpg.de). TRENDY
model output is available on request to Stephen Sitch (S.A.Sitch@exeter.ac.uk).

## Author contributions

MJ conceived the study, performed the analysis, and drafted the manuscript with intellectual input and extensive edits from all co-authors.

## Competing interests

The authors declare no competing interests.

## Acknowledgements

The authors acknowledge funding from European Space Agency Climate Change Initiative ESA-CCI RECCAP2 project (ESRIN/4000123002/18/I-NB), and EU H2020 projects, CHE (GA 776186), VERIFY (GA 776810), E-SHAPE (GA 820852), and BACI (GA 640176). We further want to thank Ana Bastos for input on an earlier version of the manuscript.

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

**Figures and Tables**

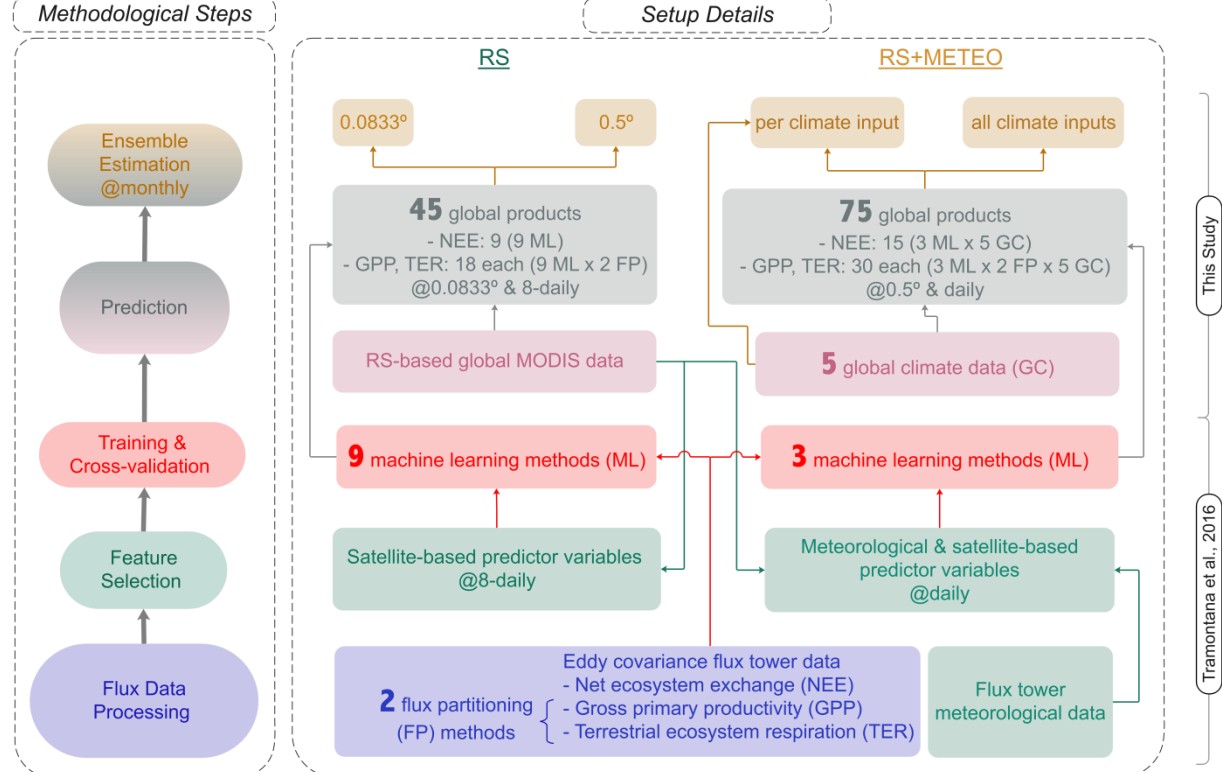

**Figure 1: Schematic overview of the methodology and data products from the FLUXCOM initiative. The flow diagram shows the methodological steps for the remote sensing -based (RS, left) and the remote sensing and meteorological data -based (RS+METEO, right) FLUXCOM products. Final monthly ensemble products for NEE, GPP, and TER from RS are available at 0.0833° and at 0.5° spatial resolution. Ensemble products from RS+METEO are available per climate forcing (GC) data set as well as a pooled ensemble at 0.5° spatial resolution. All ensemble products encompass ensemble members of different machine learning methods (ML, 9 for RS, 3 for RS+METEO) and flux partitioning methods (FP, 2 for GPP and TER).**

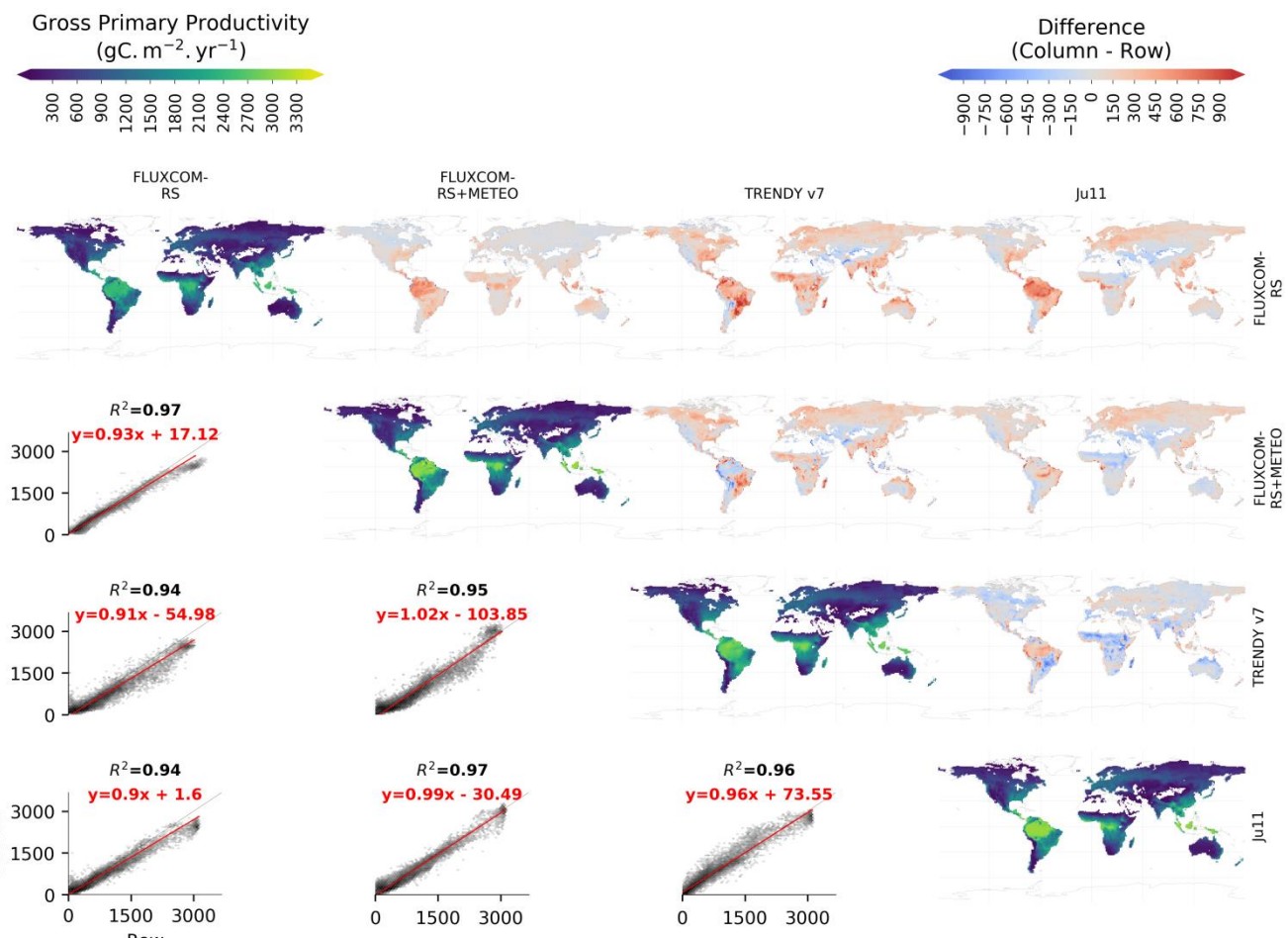


**Figure 2: Comparisons of mean annual GPP at 1° spatial resolution for the period 2008-2010 of FLUXCOM ensemble products with Ju11 and the mean of 16 TRENDY models. Diagonal: Maps of mean annual GPP. Above diagonal: Maps of GPP differences (product along column – product along row). Below diagonal: 1:1 regression where the shading shows point density. The red line and equations show the best fit line from total least square regression.**



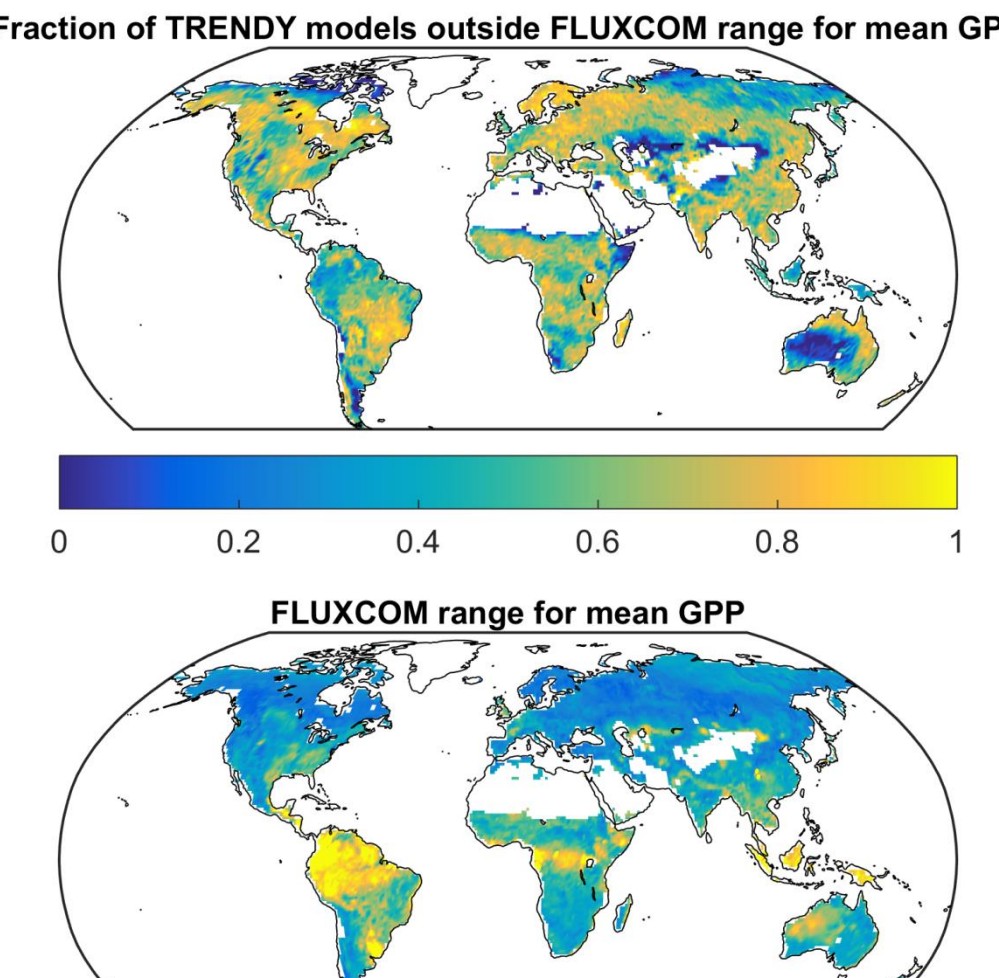


**Figure 3: Map of the fraction of TRENDY models (n=16) with mean GPP outside the range of FLUXCOM estimates.**
**The FLUXCOM range is calculated as the maximum minus minimum of all 48 FLUXCOM members from the union**
**of the RS and RS+METEO members. Mean GPP was calculated for the period 2008-2010.**

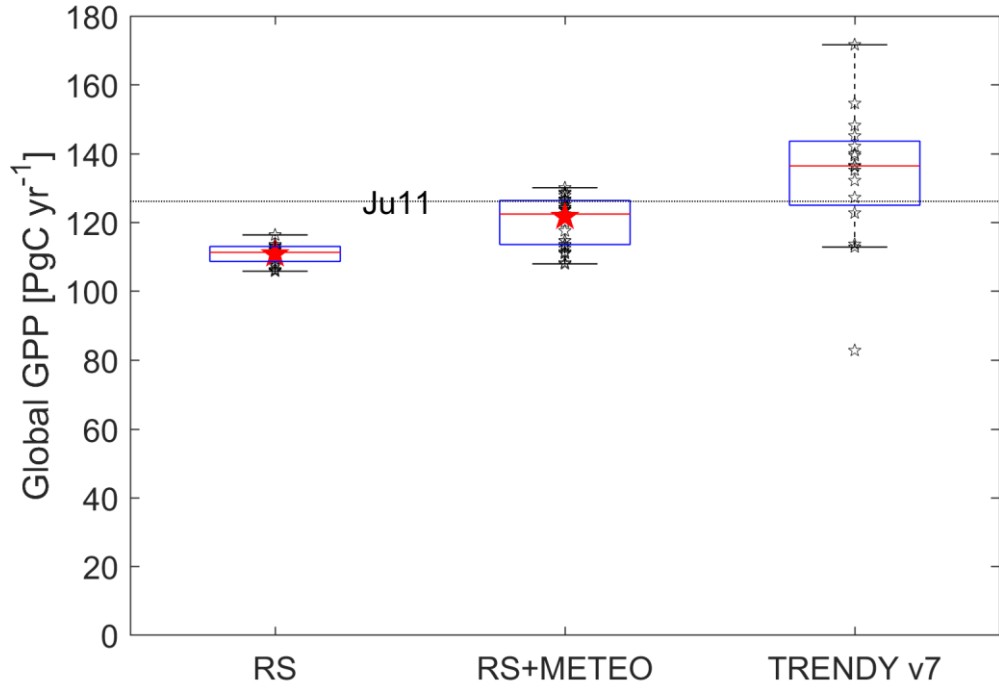


**Figure 4: Global GPP for FLUXCOM and TRENDY ensembles for the period 2008-2010. The box plots show the**
**median (red line), interquartile range (box) and total range (whiskers) of non-outliers (within median ± 1.5**
**interquartile range) of individual ensemble members (open black stars). The filled red star presents the value of the**
**ensemble product (not available for TRENDY). The estimate of Ju11 is plotted as horizontal broken line.**


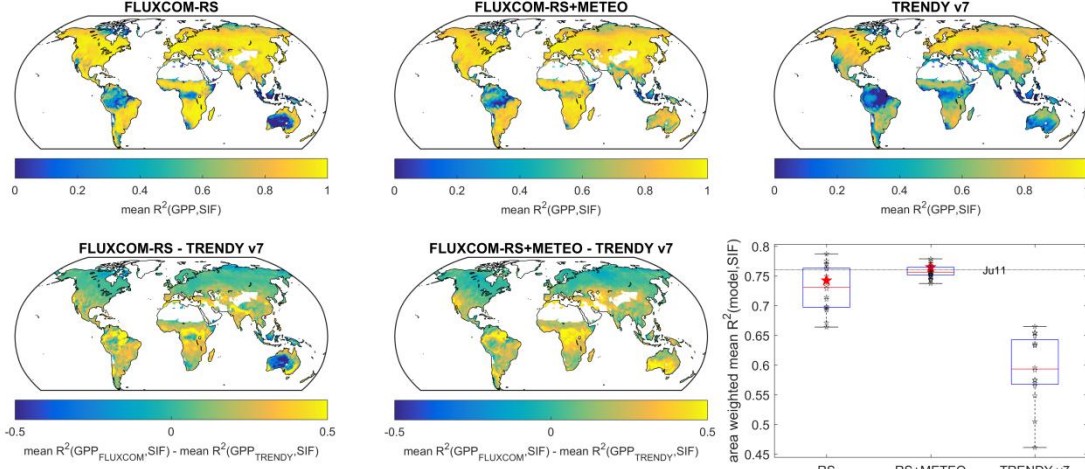


**Figure 5: Consistency of seasonal GPP variations from FLUXCOM and TRENDY with SIF from GOME-2. Maps in**
**the top row show the mean $R^2$ between mean seasonal cycles for the period 2008-2010, averaged across all respective**
**ensemble members. Difference maps in the bottom row emphasize where FLUXCOM shows better (positive value)**
**and worse (negative value) consistency with SIF than TRENDY and are based on the maps in the top row. The**
**spatially averaged $R^2$ values for the different ensembles are summarized in the bottom right panel. The box plots**
**show the distribution of individual ensemble members (open black stars). The filled red star presents the value of the**
**ensemble product (not available for TRENDY). The estimate of Ju11 is plotted as horizontal broken line.**

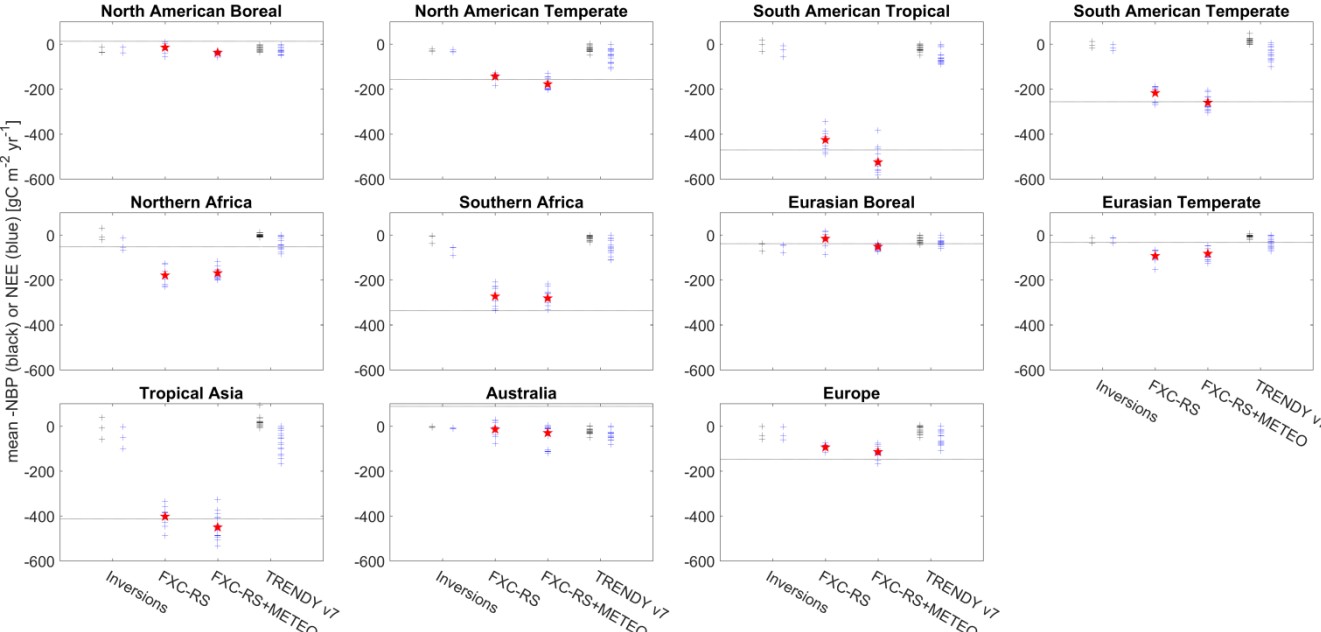


**Figure 6: Mean annual net carbon release for the years 2008-2010 over TRANSCOM regions. Crosses refer to**
**individual ensemble members where a black colour refers to negative net biome productivity (NBP, not available for**
**FLUXCOM), and blue color refers to net ecosystem exchange (NEE). For inversions, NEE was approximated by**
**correcting NBP with fire emissions (see section 2.4.3). The filled red stars refer to estimates by the ensemble product**
**of FLUXCOM setups. The horizontal broken line indicates the estimate of Ju11.**

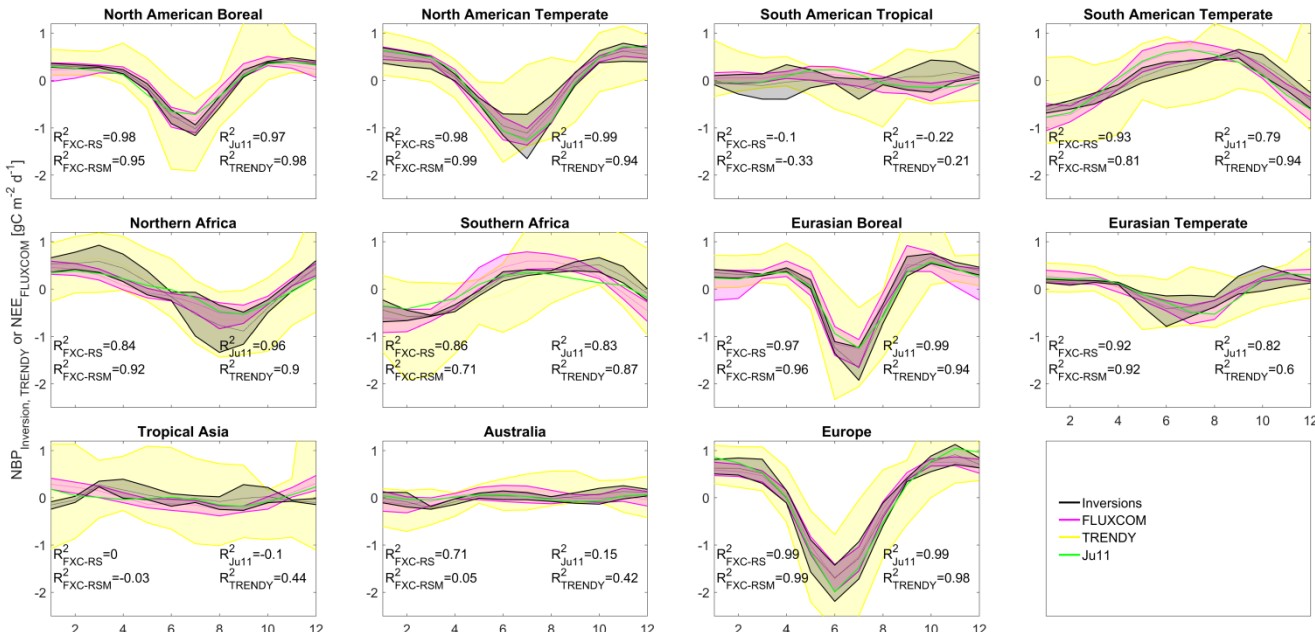


**Figure 7: Mean seasonal variations of net land carbon release for the period 2008-2010 over TRANSCOM regions. For inversions and TRENDY, -NBP was plotted, and for FLUXCOM, NEE was plotted. Please note that the region specific mean was removed for each data set. Shading indicates the range of estimates (maximum – minimum). The FLUXCOM range is based on the union of RS and RS+METEO ensemble members. $R^2$ values were calculated with the mean of the inversions. The FLUXCOM RS and RS+METEO refer to the ensemble products (median), while that for TRENDY refer to the model mean.**

1078

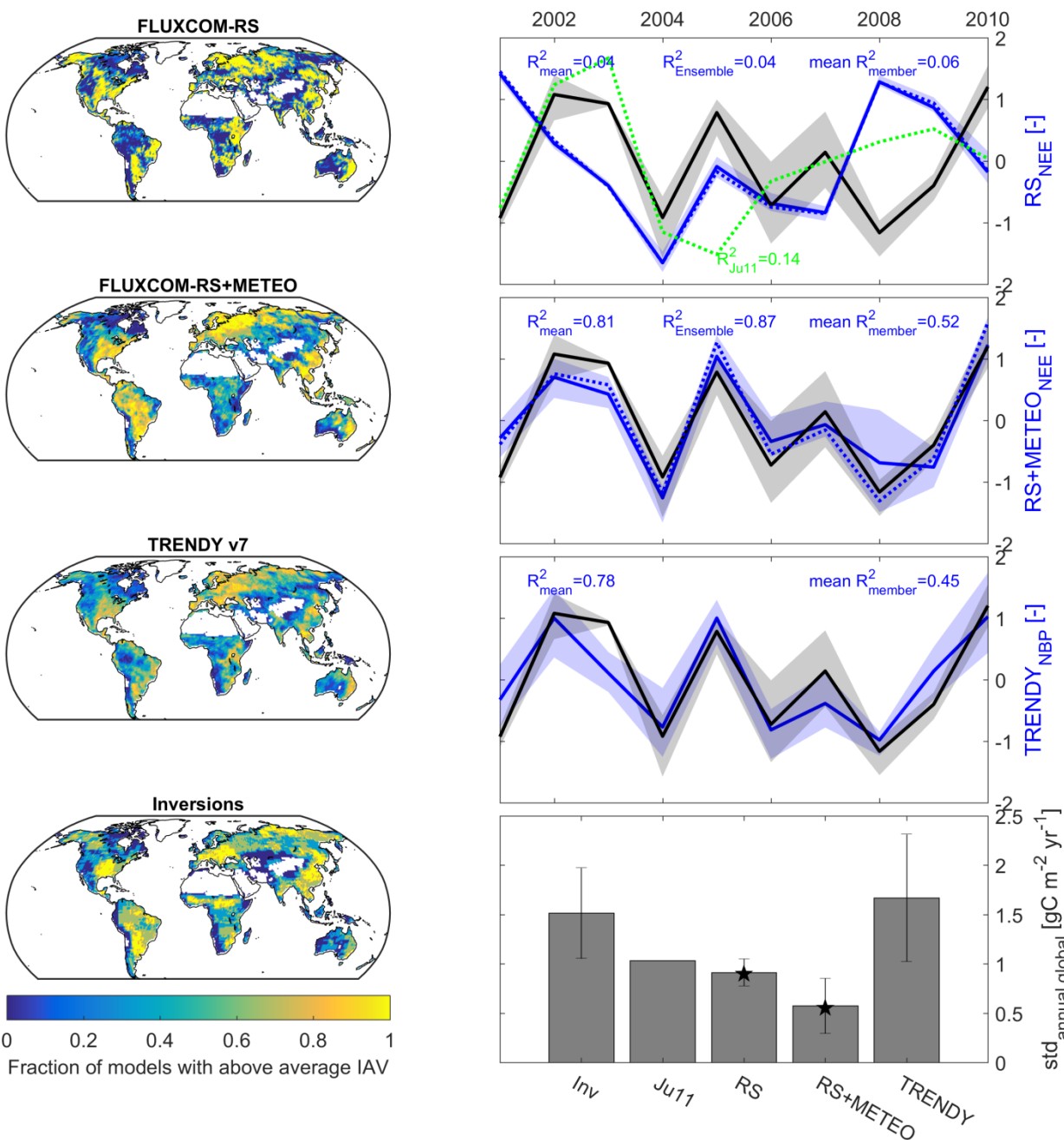

**Figure 8: Interannual variability patterns of FLUXCOM NEE, TRENDY NBP, and NBP from three atmospheric inversions for the period 2001-2010. Maps show the fraction of respective ensemble members with above average interannual variability (standard deviation of annual values multiplied with land area). Time series plots show detrended globally integrated annual NEE or NBP anomalies normalized by their standard deviation. The black line is the mean of three inversions and the gray shading indicates their range. The blue solid lines are the means of the considered ensembles; the blue dashed lines are the FLUXCOM ensemble products. $R^2$ values refer to the comparison with the mean of inversions (black solid line). The bar chart in the bottom right panel shows the standard deviation of detrended annual NEE or NBP for different data sets, averaged over the ensemble members and the error bar indicates the standard deviation of the ensemble members. Black stars for FLUXCOM refer to the value for the ensemble products.**

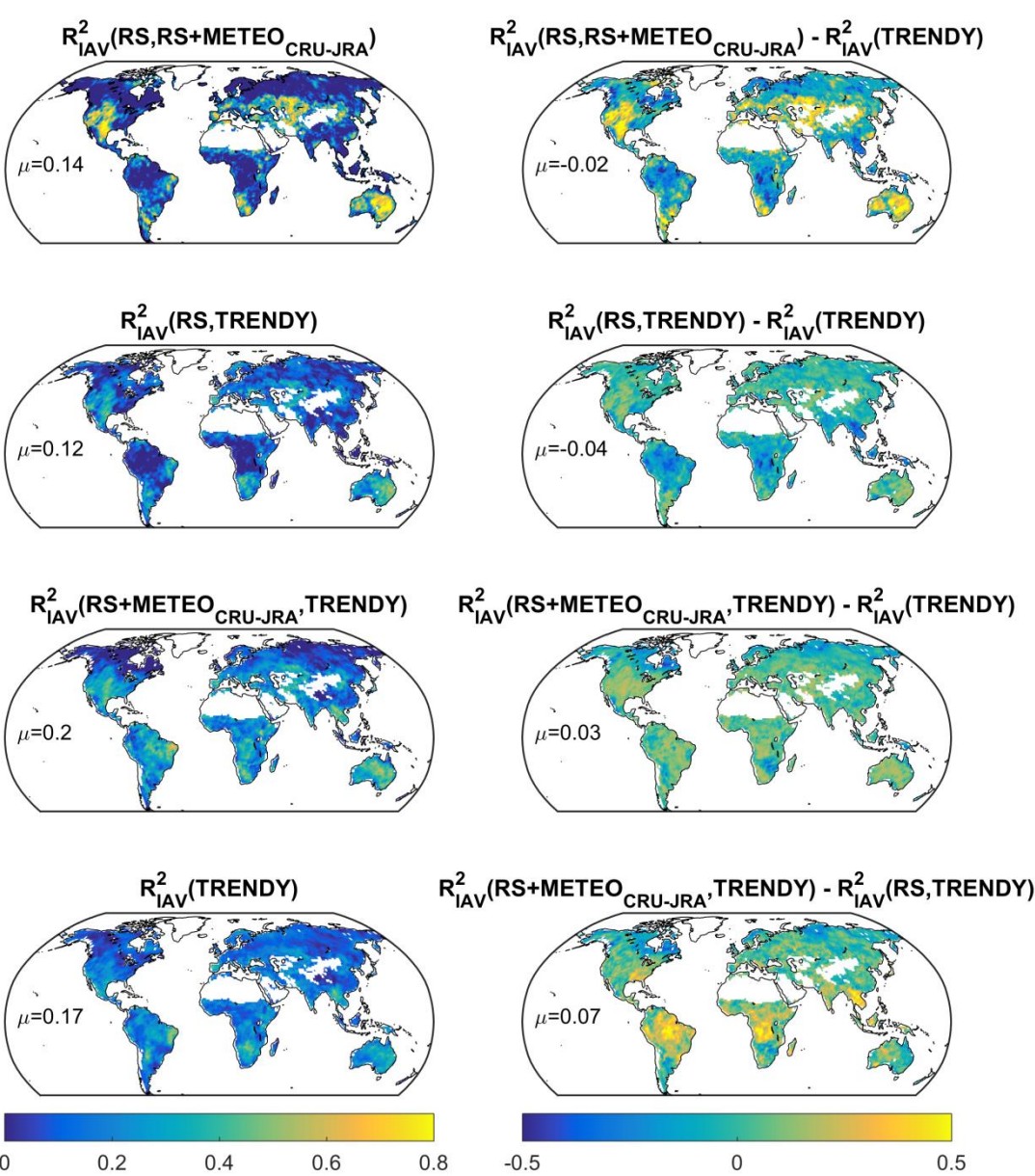


**Figure 9: Consistency between interannual variabilities (IAV) of local NEE from FLUXCOM setups and TRENDY for the period 2001-2015.**



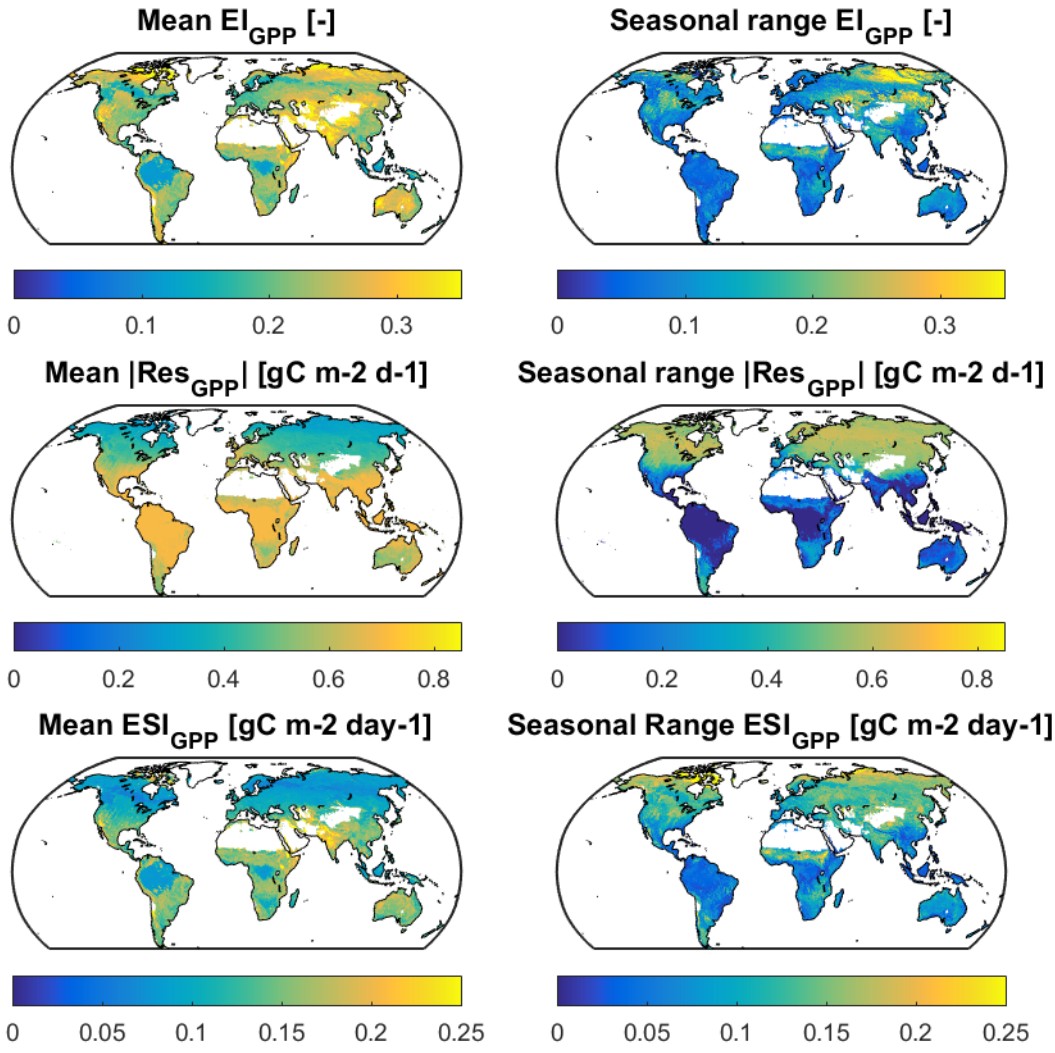


**Figure 10: Mean annual (2001-2015) and seasonal range (8-daily time step) of the Extrapolation Index (EI), the**
**expected mean absolute error of machine learning predictions, and the Extrapolation Severity Index (ESI, product of**
**the previous two) (see S2 for details) for GPP from FLUXCOM-RS.**


| Meteorological forcing data set | Spatial Resolution | Temporal Coverage |
|---|---|---|
| CRU-JRA | 0.5° x 0.5° | 1950-2017 |
| GSWP3 | 0.5° x 0.5° | 1950-2010 |
| WFDEI | 0.5° x 0.5° | 1979-2013 |
| ERA-5 | 0.5° x 0.5° | 1979-2018 |
| CERES-GPCP | 1.0° x 1.0° resampled to 0.5° x 0.5° | 2001-2013 |

**Table 1: Global meteorological forcing data sets used in FLUXCOM-RS+METEO.**



