# Peer review of "Scaling carbon fluxes from eddy covariance sites to globe: Synthesis and evaluation of the FLUXCOM approach"

_Biogeosciences, 2019_

## Referee Comment (RC1) · Anonymous Referee #1 · 6 Nov 2019

This manuscript looks at the results from an ensemble combining multiple products and machine learning algorithms to assess GPP and NEE and compare it to multiple remote sensing products. The scale of this work is truly remarkable and is clearly leading the way in combining models and machine learning algorithms, a method that will probably become more and more common.

I am not a modeler myself, but the manuscript was very detailed and easy to follow. The work was well motivated, the tests and checks were extremely thorough and well documented. The text and figures were all stellar. In particular, I found Section 4 to be particularly interesting in terms of a better understanding of what we could improve as

a community to improve the results of the models. Great work!

I realize this is somewhat beyond the scope of this manuscript, but since the machine learning algorithms are what make this work novel, it would be useful to include more details about the differences between the 9 different algorithms and what differences might be expected in the results.

Another very minor complaint is that the embedded text in many of the figures is very small and difficult to read, making it hard to figure out which panel is which. This is especially true for Figures 2,5, and 7, as well as S3 and S5.

Finally, I was surprised that Baldocchi et al 2001 was not cited since it is one of the best references regarding the FLUXNET network.

Baldocchi, D., Falge, E., Gu, L., Olson, R., Hollinger, D., Running, S., . . . Wofsy, S. (2001). FLUXNET: A New Tool to Study the Temporal and Spatial Variability of Ecosystem-Scale Carbon Dioxide, Water Vapor, and Energy Flux Densities. Bulletin of the American Meteorological Society, 82(11), 2415–2434. https://doi.org/10.1175/1520-0477(2001)082<2415:FANTTS>2.3.CO;2

---

## Referee Comment (RC2) · Anonymous Referee #2 · 22 Dec 2019

I found the manuscript to be interesting and appropriately self-critical with good uncertainty accounting. Any comments that I have would not improve the manuscript appreciably but I do suggest one more careful read for minor (see e.g. line 360, 'poorly') usage issues that may or may not be caught during copyediting.

---

## Author Comment (AC2) · 29 Jan 2020

We thank reviewer #2 for the positive assessment of our work and manuscript.

We reformulated the mentioned sentence to:

"Here, the Random Forests method performed less well compared to the other two methods."

We also carefully checked appropriate usage of terms in the manuscript and revised accordingly if needed.

---

## Author Comment (AC3) · 29 Jan 2020

We will carefully revise and improve all relevant figures according to the reviewers suggestion.

We have included the Baldocchi et al. 2001 reference in the first sentence of the introduction in the revised manuscript as we had missed this reference.

---

## Author Response (AR1)

**Author response to reviewer comments**

Reviewer #1

This manuscript looks at the results from an ensemble combining multiple products and machine learning algorithms to assess GPP and NEE and compare it to multiple remote sensing products. The scale of this work is truly remarkable and is clearly leading the way in combining models and machine learning algorithms, a method that will probably become more and more common. I am not a modeler myself, but the manuscript was very detailed and easy to follow. The work was well motivated, the tests and checks were extremely thorough and well documented. The text and figures were all stellar. In particular, I found Section 4 to be particularly interesting in terms of a better understanding of what we could improve as a community to improve the results of the models. Great work!

Thank you!

I realize this is somewhat beyond the scope of this manuscript, but since the machine learning algorithms are what make this work novel, it would be useful to include more details about the differences between the 9 different algorithms and what differences might be expected in the results.

Reviewer #1 raises a challenging point here as machine learning methods differ fundamentally in their maths, algorithms, and approaches to training and hyper-parameter tuning. This has been described in the FLUXCOM cross-validation paper by Tramontana et al. 2016 with ample references therein for further information. It is very hard, probably impossible, to anticipate how results would differ by machine learning choice. Therefore, different methods have been included in the FLUXCOM ensemble. One overall conclusion of our synthesis is that the choice of the predictor set given to the machine learning methods matters more than the choice of machine learning method. Due to this finding, results differing by machine learning method are presented in the supplementary material rather than the main article. We therefore agree with the reviewer that including more details on machine learning methods is beyond the scope of the paper and would not improve clarity.

Another very minor complaint is that the embedded text in many of the figures is very small and difficult to read, making it hard to figure out which panel is which. This is especially true for Figures 2,5, and 7, as well as S3 and S5.

Thank you for this comment. We will carefully revise and improve all relevant figures accordingly.

Finally, I was surprised that Baldocchi et al 2001 was not cited since it is one of the best references regarding the FLUXNET network.

We thank reviewer #1 for this catch. We have included the Baldocchi et al. 2001 reference in the first sentence of the introduction in the revised manuscript.

Reviewer #2

I found the manuscript to be interesting and appropriately self-critical with good uncertainty accounting. Any comments that I have would not improve the manuscript appreciably but I do suggest one more careful read for minor (see e.g. line 360, 'poorly') usage issues that may or may not be caught during copyediting.

Thank you! We reformulated the mentioned sentence to:

*"Here, the Random Forests method performed less well compared to the other two methods."*

We also carefully checked appropriate usage of terms in the manuscript and revised accordingly if needed.

[revised manuscript text omitted]